# Theoretical Analysis and Verification on Plastic Deformation Behavior of Rocket Nozzle Using a Novel Tube Upsetting-Bulging Method

**DOI:** 10.3390/ma16041680

**Published:** 2023-02-17

**Authors:** Yizhe Chen, Shilong Zhao, Hui Wang, Jun Li, Lin Hua

**Affiliations:** 1Hubei Key Laboratory of Advanced Technology for Automotive Components, Wuhan University of Technology, Wuhan 430070, China; 2Hubei Collaborative Innovation Center for Automotive Components Technology, Wuhan 430070, China; 3Jiangsu Xinyang New Material Co., Ltd., Yangzhou 225000, China; 4Hubei Engineering Research Center for Green & Precision Material Forming, Wuhan 430070, China

**Keywords:** rocket nozzle, tube upsetting-bulging, plastic deformation, critical pressure

## Abstract

The rocket nozzle is one of the core components to ensure the safe flight of rockets. To overcome the problems of multi-step forming, the occurrence of defects, and severe plastic deformation in traditional technology, a novel forming method named tube upsetting-bulging (TUBG) is put forward. With the support of internal pressure, a tube is deformed with an upsetting and bulging process at the same time. The tube is thickened at the small end and thinned at the large end. A nozzle with sharply varying diameters can be obtained. A theoretical model of TUBG that considers wrinkles and rupture is built. The influence factors of internal pressure during TUBG are discussed. Experiments and simulation works are conducted to analyze the plastic deformation process of TUBG. Results show that mechanical properties and geometrical parameters have an obvious influence on critical internal pressure. The proposed theoretical model can be used to predict a forming zone without wrinkles, rupture, and severe strain values. A well-formed nozzle can be obtained using the predicted forming zone, which verifies the correctness of the theoretical analysis. It can be found that TUBG is a novel potential method to fabricate rocket nozzles with high efficiency and quality without defects.

## 1. Introduction

The rocket nozzle is the key device and power source of the rocket engine [1]. It converts the chemical energy generated by a propellant into kinetic energy, which can provide the driving force for the rocket. Therefore, the rocket nozzle is extremely important to the safe flight of the rocket [2,3,4]. The rocket nozzle is subjected to 6–10 Mpa pressure and high temperature (2800–3400 K) gas erosion when working [5,6,7].

In previous studies, the spinning process is widely used to make rocket nozzles [8,9]. Spinning is a method that uses the movement of a rotary wheel to press the blank rotating with a mold. It causes continuous plastic deformation of metals and finally obtains the hollow rotation parts [10]. However, with the requirements of lightweight and high strength, spinning still has the following disadvantages: 1. Spinning requires multi-step forming. When a rocket nozzle is manufactured, the spinning process needs to be carried out two or more times [11]. 2. For parts with thin wall thickness, it is easy to wrinkle and rupture at the same time. 3. The degree of plastic deformation is large. Due to the reciprocating action of the rotary wheel, the microstructure of the rocket nozzle is elongated. Therefore, the grain is easy to grow up abnormally after heating, which does not meet the requirements under high temperatures and pressure [10]. With the increasing difficulty of space exploration missions, the requirements for the rocket nozzle are becoming significantly stricter. The abovementioned disadvantages in the traditional methods have limited the development of the rocket nozzles inevitably.

The light weight of aerospace and vehicle-carrying equipment has been attracting attention [12,13]. For thin-wall hollow structural components, a common forming process is high-pressure tube hydroforming (HPTH) [14,15,16]. In this method, internal pressure is applied to force the tube to expand into the required shape in a closed die cavity −much like blowing up a balloon [16]. A tube blank of the initial type selected for manufacture must have less than the circumference of the lowest section of the product to use HPTH. However, internal pressure is the main and only driving force in the HPTH. When the size of tube deformation is large, pressures of hundreds of Mpa are needed [17]. In addition, due to the influence of friction caused by high pressure, the material of the tube has difficultly flowing, which can cause serious thickness thinning or even cracking [18,19]. As a result of the limitation of the ultimate expansion coefficient of the tube, the maximum expansion rate of the tube section is generally 20–33% [20].

Another method to make the thin-wall hollow structural components is the tube upsetting [21,22]. It can reduce the diameter of the tube through the mold [23]. Obviously, in order to manufacture the required parts by tube upsetting, the circumference of the selected initial tube blank must be greater than the maximum section of the product. Therefore, when the circumference of the tube decreases, wrinkles are always the main defects in the tube upsetting [24,25]. In particular, for the thin wall hollow tube without internal support, when a large degree of tube upsetting deformation occurred, it is more likely to appear as dead wrinkles, which leads to the failure of forming.

The tube-forming methods mentioned above have certain limitations (wrinkles that cannot be suppressed, bursting risks under the action of high pressure). Therefore, Chu Guannan et al. proposed a new method called tube hydro-forging [26]. Under the support of internal pressure, the stamping of the outer mold is used to shape the tube into the desired shape. In this scheme, internal pressure only plays a supporting role. The pressure required for forming is reduced to 30% of the hydraulic forming. As a result, the use of high pressure is avoided, and the wrinkles of the tube are suppressed. However, the above forming schemes are all developed for the square tube members with the small size of the vehicle. When the upsetting rate of the cross-section reaches more than 56%, this scheme is limited greatly. In addition, when the thickness of the part is reduced to less than 0.5 mm, the tube can easily lose stability and wrinkle.

Especially for rocket nozzles, as a thin-wall annular component, which has a sharply varying diameter from one end to the other end. It is thickened at the small end and thinned at the large end. When a tube with a smaller diameter is used in the manufacture of a rocket nozzle, it carries the risk of rupture depending on high-pressure tube hydroforming (HPTH). When a tube with a larger diameter is used in the manufacture of a rocket nozzle, there is still a risk of wrinkles, even with the support of internal pressure, because of the large degree of deformation.

For a long time, studies on the deformation of thin-wall tubes only stayed on the single deformation (stamping, upsetting, bulging, hydroforming), the case of combining upsetting and bulging simultaneously has not been examined or reported. For this purpose, we proposed a new tube-forming method called tube upsetting-bulging (TUBG). Different from the traditional methods, the deformation of upsetting and bulging occur on the different parts of the same tube in TUBG. A theoretical model of TUBG that considers wrinkling and rupturing is built. The influence factors of internal pressure during TUBG are discussed. Experiments and simulation works are conducted to analyze the plastic deformation process of TUBG.

## 2. Principle of Tube Upsetting-Bulging Method

### 2.1. Forming Processes

The schematic diagram of the TUBG studied in this paper is shown in Figure 1. For the rocket nozzle, as shown in Figure 1a, the small end is thicker, and the large end is thinner. Compared with the high-pressure tube hydroforming or tube upsetting, during TUBG, instead of a single bulging or upsetting, the different parts of the tube are upset and expanded, respectively. In the process of tube upsetting-bulging (TUBG), by controlling the movement of the punch, the tube is deformed under the force of the punch and mold. By operating the pump, the pressure provided by the liquid drives the tube to have plastic deformation. As a result, one end of the tube is upsetting, and the other end is bulging. The wall thickness of the smaller end increases and the wall thickness of the larger end decreases because of the same volume of the metals. Combining the above processes, the thickness of the smaller end and larger end of the nozzle is controlled. In this scheme, since one end of the tube is upsetting and the other end is bulging, we call this process tube upsetting-bulging.

Taking a certain type of rocket engine nozzle as an example, the diameter ratio of the small end to the big end is 31.67%. The wall thickness of the small end is thicker, and the wall thickness of the large end is thinner. According to the principle of upsetting-bulging, it is advisable to use an appropriate diameter in the middle of the nozzle as the diameter of the original tube. During TUBG, the tube is supported by a rubber container with liquids, which can provide internal pressure. The upper half of the tube meets the punch of the stamping machine, it is knocked into the cavity of the mold under the action of the punch until it is completely fitted. In this process, the perimeter of the tube decreases, the section of the tube is reduced, and the wall thickness increases. Subsequently, the internal pressure is increased. The cross-section is expanded, and the wall thickness is thinned until it fits into the mold.

### 2.2. Advantages of Novel Method

From the principle of TUBG, the tube only has soft contact with the rubber container in the whole process of forming. There is no reciprocating effect of the rigid wheel. Compared with the traditional spinning forming, the product of TUBG has excellent surface properties. It will be conducive to the coating of the thermal insulation layer on the surface of the rocket nozzle, which improves the reliability of the nozzle in the service process effectively. At the same time, to meet the size requirements of the target parts, the upsetting and bulging sections can be formed at one time. Therefore, it shortens the production cycle and improves the qualification rate, which also ensures the dimensional accuracy better.

Compared with the traditional tube forming methods, in the process of TUBG, the deformation force driving the upsetting is provided by the stamping machine. Internal pressure only plays a supporting role. It greatly reduces the demand for internal high pressure. Therefore, the influence of friction caused by high pressure is weakened. At the same time, upsetting and bulging occur in different parts of a single tube, which divides the traditional severe plastic deformation into two parts. The forming quality of the product is better improved. In addition, because two opposite deformation behaviors occur on a single tube in one process, TUBG can be used to manufacture components with large variations in diameter from small to large ends.

It can be seen that if the internal pressure is too low during upsetting, the tube has the tendency to wrinkle. If the internal pressure is too high during bulging, the tube has the tendency to rupture. These are two typical geometric defects in the upsetting-bulging process. Therefore, according to the size and properties of the materials, it is important to understand the influencing factors of wrinkling and rupturing. On this basis, the minimum pressure to eliminate wrinkling and the maximum pressure to prevent rupturing can be determined. Further, the forming zone which meets the forming needs will be established. To achieve this the corresponding analytical model will be proposed in the next section.

## 3. Theoretical Analysis

### 3.1. Theoretical Model

As shown in Figure 2, during TUBG, it is important to determine the critical minimum support pressure to prevent wrinkling and the critical maximum rupture pressure to prevent breaking. Therefore, it is necessary to propose a TUBG theoretical model with a correction coefficient. Subsequently, the relative forming zone, according to the theoretical calculation, can be determined.

#### 3.1.1. Mechanism of Wrinkles Suppression

When upsetting occurs, due to the incompressibility of the metals, the perimeter of the tube section decreases, and the thickness increases. For wrinkling cases, it is assumed that the circumference of the round tube remains unchanged [27]. When the internal pressure is lower than the critical value, the cross-section of the wrinkled tube is shown in Figure 3b. With internal pressure support, the perfect upsetting tube is shown in Figure 3c.

When the upsetting process is completed, there are two results: the tube keeps perfect or it wrinkles. The appearance of wrinkles can be regarded as the behavior to release the tube’s internal energy. The strain energy *E*_1_ contained in the unwrinkled tube is greater than the strain energy *E*_2_ contained in the wrinkled tube. The energy released by wrinkling can be written as *E*_0_. The relationship exists as follows:(1)$$E_{0}=E_{1}-E_{2}$$

Under the action of critical internal pressure *P_L_*, when the tube with a wrinkled area of *A* is suppressed, the energy can be written as:(2)$$E_{0}=P_{L}\int dA=P_{L}A$$

The process of inhibiting wrinkles can be seen as a force *F,* forcing the wrinkles to move a distance. When studying the wrinkle behavior of the thin plate, Cao et al. [28] deduced the critical internal pressure as:(3)$$P_{L}=\frac{3\left(E_{1}-E_{2}\right)}{4\varphi s}$$

φ is the height of wrinkles and *s* is the length of wrinkles.

In the past, when scholars studied the upsetting behavior of tubes, the shape of the wrinkled part *y* was regarded as a sine wave [26,27], which can be expressed as a function with *x* as an intermediate variable:(4)$$y=\frac{\varphi }{2}\left(1-\cos ax\right)$$

Assuming without the internal pressure, when the displacement of *u*_1_ occurs at the edge of the wrinkle, the frequency *a* of the corresponding mode can be expressed as:(5)$$a=\frac{2\pi }{s-2\Delta u_{1}}$$

In the work of previous scholars [26], *s* – 2Δ*u*_1_ can be expressed as $se^{\varepsilon_{1}}$. Therefore, the height φ of wrinkles can be expressed as:(6)$$\varphi =\frac{2y}{1-\cos\frac{2\pi }{se^{\varepsilon_{1}}}x}$$

The length of the wrinkles can be expressed according to the means of calculus as:(7)$$s=\int_{0}^{s-\Delta u_{1}}\sqrt{1+y^{2}}dx$$

Consequently, φ can be expressed as:(8)$$\varphi =\frac{2s}{\pi }\sqrt{e^{\varepsilon_{1}}-e^{2\varepsilon_{1}}}$$

When studying the wrinkling behavior, Cao et al. [28] deduced the strain energy contained in the unwrinkled tube as follows:(9)$$E_{1}=\frac{sKt}{n+1}{\left(\varepsilon_{0}-\frac{2\sqrt{3}}{3}\varepsilon_{1}\right)}^{n+1}$$

$\varepsilon_{0}$ is the pre-strain. In this experiment, the pre-strain of the tube without deformation can be regarded as 0.

There is a relationship between the stress and strain of the material as follows:(10)$$\sigma =K\varepsilon^{n}$$
where *K* is the strength coefficient, and *n* is the strain-hardening exponent. *σ* and *ε* are effective values.

After upsetting, the shape of the tube from the perspective of the XOY plane is shown in Figure 4.

During the tube upsetting process, it meets the simple loading conditions. From the full volume theory, the correlation between *ε* and *σ* can be rewritten as:(11)$$\frac{\varepsilon_{r}^{’}}{\sigma_{r}^{’}}=\frac{\varepsilon_{\theta }^{’}}{\sigma_{\theta }^{’}}=\frac{\varepsilon_{t}^{’}}{\sigma_{t}^{’}}$$

Using the law of equal proportionality, the above equation can be rewritten as:(12)$$\frac{\varepsilon_{r}^{’}-\varepsilon_{\theta }^{’}}{\sigma_{r}^{’}-\sigma_{\theta }^{’}}=\frac{\varepsilon_{\theta }^{’}-\varepsilon_{t}^{’}}{\sigma_{\theta }^{’}-\sigma_{t}^{’}}$$

According to the theory of thin plate and the theory of thin-wall shell [29,30], since the diameter is much larger than the thickness, the principal stress *σ_t_* perpendicular to the direction of the material surface is considered 0, the above equation can be regarded as:(13)$$\frac{\varepsilon_{r}^{}-\varepsilon_{\theta }^{}}{\sigma_{r}^{}-\sigma_{\theta }^{}}=\frac{\varepsilon_{\theta }^{}-\varepsilon_{t}^{}}{\sigma_{\theta }^{}}$$

Record the angle between the contour of the mold and the axis as α, $\varepsilon_{t}$ can be written as:(14)$$\varepsilon_{t}=-\frac{\sigma_{\theta }+\sigma_{r}}{2\sigma_{\theta }-\sigma_{r}}\varepsilon_{\theta }=-\frac{1+(1+\mu \cot\alpha )(1-\frac{r}{R})}{2-(1+\mu \cot\alpha )(1-\frac{r}{R})}\ln\frac{R}{R_{0}}$$

For tubes with stable upsetting, there is an equilibrium equation:(15)$$R\frac{d\sigma_{r}}{dR}+\sigma_{r}-\sigma_{\theta }(1+\mu \cot\alpha )=0$$

In the upsetting area, the compressive stress $\sigma_{\theta }$ can be expressed as:(16)$$\sigma_{\theta }=\beta Y$$

β is a constant. In the study of this forming process, 1.15 can be taken.

Because the degree of deformation varies from point to point within the deformation area, the true stress Y can be expressed as:(17)$$Y=\sigma_{s}+(\frac{Y_{b}-\sigma_{s}}{\ln\sigma_{b}})(1-\frac{R}{R_{0}})$$

In summary, the equilibrium equation can be written as:(18)$$R\frac{d\sigma_{r}}{dR}+\sigma_{r}-1.15\left[\sigma_{s}+(\frac{Y_{b}-\sigma_{s}}{\ln\sigma_{b}})(1-\frac{R}{R_{0}})\right](1+\mu \cot\alpha )=0$$

Integrate the above equation:(19)$$R\sigma_{r}=1.15(1+\mu \cot\alpha )\left[\sigma_{s}R+R(\frac{Y_{b}-\sigma_{s}}{\ln\sigma_{b}})-(\frac{Y_{b}-\sigma_{s}}{\ln\sigma_{b}})\frac{R^{2}}{2R_{0}}\right]+C$$

When R=r, $\sigma_{r}=0$, C can be written as:(20)$$C=-1.15(1+\mu \cot\alpha )\left[\sigma_{s}r+r(\frac{Y_{b}-\sigma_{s}}{\ln\sigma_{b}})-(\frac{Y_{b}-\sigma_{s}}{\ln\sigma_{b}})\frac{R^{2}}{2R_{0}}\right]$$

The distribution law of $\sigma_{r}$ in the deformation zone of the workpiece can be written as:(21)$$\sigma_{r}=1.15(1+\mu \cot\alpha )(1-\frac{r}{R})\left[\sigma_{s}+(\frac{Y_{b}-\sigma_{s}}{\ln\sigma_{b}})(1-\frac{Rr}{2R_{0}})\right]$$

At the same time, under the condition that the material is not compressed, Cao [28] et al. provide the formula for calculating the strain energy contained in the wrinkled tube:(22)$$E_{2}=\frac{2Kt}{n+1}{\left(\frac{\sqrt{3}t}{3}\right)}^{n+1}{\left(\frac{2}{\varphi^{2}\Delta a}+\frac{t}{2}\right)}^{-n}\tan^{-1}\left(\frac{\varphi \Delta a}{2}\right)$$

It can be observed that the wrinkle length s and the tube wall thickness t are the same dimensions, so there are multiple relationships between the defined wrinkle length and the tube wall thickness:(23)$$N=\frac{s}{t}$$

Combined with the above formula, the critical internal pressure *P_cr_*_1_ of the tube without wrinkling can be expressed as:(24)$$p_{c_{1}}=\frac{3K\pi }{4N\sqrt{e^{\varepsilon_{1}}-e^{2\varepsilon_{1}}}}\left[\varepsilon_{1}{}^{n+1}-{\left(\frac{\sqrt{3}t}{3}\right)}^{n}\left(\frac{Nte^{2\varepsilon_{1}}}{4\sqrt{e^{\varepsilon_{1}}-e^{2\varepsilon_{1}}}}+\frac{t}{2}\right)\left(\frac{2}{N}\right)\tan^{-1}\left(\frac{2\sqrt{e^{\varepsilon_{1}}-e^{2\varepsilon_{1}}}}{e^{\varepsilon_{1}}}\right)\right]$$

#### 3.1.2. Mechanism of Resistance to Rupturing

After bulging, the shape of the tube from the perspective of the XOY plane is shown in Figure 5. For the bulging phase, it is generally assumed that the stress-strain is evenly distributed along the wall thickness. The radial stress is ignored. Only the tangential tensile stress is considered.

Under the drive of internal pressure, the tube is expanded to the maximum diameter. Taking a unit width of the tube for analysis, according to the equilibrium conditions of the round tube, it can be obtained that:(25)$$\int_{0}^{x}pr\sin\alpha d\alpha =2\sigma_{t}t$$

It can be simplified as:(26)$$p=\frac{t}{r}\sigma_{t}$$

When studying the ultimate expansion coefficient of the tube, considering the thick anisotropy of the material, Yang [31] et al. provide the influence coefficient of the above formula:(27)$$M=\frac{1+R}{\sqrt{1+2R}}$$

*R* is the anisotropic parameter of the selected material.

Considering the influence of cold work hardening of the material, replace $\sigma_{t}$ with $\sigma_{b}$, combined with the influence coefficient, and the critical pressure required for expansion can be corrected to:(28)$$p=\frac{1+R}{\sqrt{1+2R}}\frac{t}{r}\sigma_{b}$$

Assuming that the ratio of axial stress to circumferential stress is ξ, the initial critical yield pressure obtained from the Tresca yield criterion is:(29)$$p_{s}=\frac{1}{1-\xi }\frac{t}{r}\sigma_{s}$$

During the bulging process, there are:(30)$$\left\{ \begin{array}{l} \sigma_{z}=\ln\left(\frac{l}{L_{0}}\right)\\ \sigma_{\theta }=\ln\left(\frac{r}{R_{0}}\right) \end{array}\right.$$

Combined with the above influence coefficients, when plastic deformation occurs, the ring stress and axial stress are derived as:(31)$$\left\{ \begin{array}{l} \sigma_{\theta }=\frac{{\left(1+R\right)}^{2}}{1+2R}\frac{\sigma_{e}}{\varepsilon_{e}}\left(\frac{R}{1+R}\varepsilon_{z}+\varepsilon_{\theta }\right)\\ \sigma_{z}=\frac{{\left(1+R\right)}^{2}}{1+2R}\frac{\sigma_{e}}{\varepsilon_{e}}\left(\frac{R}{1+R}\varepsilon_{x}+\varepsilon_{z}\right) \end{array}\right.$$

To deform the tube without failure conditions such as rupture, Chow [32] et al. provide a critical pressure including the thick anisotropy parameter when the tube is expanded:(32)$$p_{c_{2}}=K\frac{t_{0}}{r_{0}e^{n}}\frac{1+R}{\sqrt{1+2R}}{\left(\frac{n}{2}\frac{1+R}{\sqrt{1+2R}}\right)}^{n}$$

In summary, the selectable pressures $p_{cr}$ for this process are between $p_{cr_{1}}$ and $p_{cr_{2}}$:(33)$$\frac{3K\pi }{4N\sqrt{e^{\varepsilon_{1}}-e^{2\varepsilon_{1}}}}\left[\varepsilon_{1}{}^{n+1}-{\left(\frac{\sqrt{3}t}{3}\right)}^{n}\left(\frac{Nte^{2\varepsilon_{1}}}{4\sqrt{e^{\varepsilon_{1}}-e^{2\varepsilon_{1}}}}+\frac{t}{2}\right)\left(\frac{2}{N}\right)\tan^{-1}\left(\frac{2\sqrt{e^{\varepsilon_{1}}-e^{2\varepsilon_{1}}}}{e^{\varepsilon_{1}}}\right)\right]\leq p_{cr}\leq K\frac{t_{0}}{r_{0}e^{n}}\frac{1+R}{\sqrt{1+2R}}{\left(\frac{n}{2}\frac{1+R}{\sqrt{1+2R}}\right)}^{n}$$

### 3.2. Influencing Factors of Pressure

#### 3.2.1. Materials

The proposal in this process is derived from two hypotheses: the length of wrinkles gradually decreases until it disappears with the increase of internal pressure; the risk of rupture gradually decreases until it disappears with the decrease of internal pressure. The experimental results confirm the above two hypotheses.

Figure 6 shows the critical wrinkling pressure of 20 steel and AA6061. For 20 steel, with the increase of internal pressure, the wrinkles can be suppressed. When the internal pressure reaches 10.4 MPa, the wrinkles eventually disappear. That means that the critical wrinkling pressure of a 20 steel tube with a punch stroke of 100 mm is 10.4 MPa. When the internal pressure is higher than 10.4 MPa, the 20 steel tube can achieve stable upsetting deformation. For AA6061, the critical wrinkling pressure with a punch stroke of 100 mm is 7.9 MPa. Compared to 20 steel, the critical wrinkling pressure of AA6061 is 24% lower.

Figure 7 shows the critical rupture pressure of 20 steel and AA6061. It is clear from the picture that for tubes that have undergone bulging behavior, the risk of tube rupture can be reduced with the reduction of internal pressure. When the internal pressure is reduced to 13.9 MPa, for the 20 steel tube with a radius of 75 mm and a wall thickness of 2 mm, the rupture defect eventually disappears. That means the critical rupture pressure is 13.9 MPa. When the internal pressure is lower than 13.9 MPa, the 20 steel tube can produce mold bugling behavior without rupture. For AA6061, the critical rupture pressure is 7.2 MPa. Compared to 20 steel, the critical rupture pressure of AA6061 is 48.2% lower.

As shown in Figure 8, the TUBG forming zone of 20 steel can be obtained after combing the above critical pressure.

#### 3.2.2. Mechanical Properties

In this paper, the critical rupture pressure is obtained for different strength coefficients, strain hardening exponent, and anisotropy. As shown in Figure 9, when the anisotropy increases from 0.68 to 1.28, the critical rupture pressure gradually increases. For tubes with a diameter of 60 mm, when the anisotropy increases from 0.68 to 1.28, the critical rupture pressure increases from 17.3 MPa to 19.4 MPa, with an increasing range of 12.1%. For tubes with a diameter of 80mm, when the anisotropy increases from 0.68 to 1.28, the critical rupture pressure increases from 12.9 MPa to 14.1 MPa, with an increasing range of 9.3%.

As shown in Figure 10, when the strain hardening exponent increases from 0.15–0.30, the critical internal pressure gradually decreases. When 0 < *ε* < 1 and *K* are given, the strain hardening exponent n of the material decreases, and the value of σ is larger, which means the material has higher strength and is difficult to rupture during the bulging phase. For tubes with a diameter of 60 mm, when the strain hardening exponent increases from 0.15–0.30, the critical rupture pressure decreases from 14.7 MPa to 20.2 MPa, with a decreasing range of 37.4%. For tubes with a diameter of 80 mm, when the strain hardening exponent increases from 0.15–0.30, the critical rupture pressure increases from 11.1 MPa to 15.2 MPa, with a decreasing range of 36.9%.

As shown in Figure 11, when the strength coefficient increases from 700–1000 MPa, the critical rupture pressure gradually increases. For tubes with a diameter of 60 mm, when the strength coefficient increases from 700–1000 MPa, the critical rupture pressure increases from 13.9 MPa to 20.0 MPa, with an increasing range of 43.8%. For tubes with a diameter of 80 mm, when the strength coefficient increases from 700–1000 MPa, the critical rupture pressure increases from 10.5 MPa to 14.9 MPa, with an increasing range of 41.9%. Strength coefficient *K* describes the strength of the materials. The material with a higher value of *K* has higher strength, which means that it is difficult to rupture during the bulging phase.

As shown in Figure 12, it compares the critical rupture pressure between 20 steel and AA6061 using TUBG. To meet the needs of this process, materials with large strength coefficients, anisotropy, and a small hardening index should be selected. That means a larger forming zone can be obtained. When the above parameters change, these rules and equations are still valid for other materials.

#### 3.2.3. Geometric Parameters

This paper obtains the forming zone for tubes with different thickness-diameter ratios under the condition of a 100 mm stamping stroke. In this work, the mold is unchanged. Therefore, the outer contour of the nozzle is fixed. The smaller thickness-diameter ratio means the tube has a smaller wall thickness. When the thickness-diameter ratio is reduced from 1.67% to 1%, the thickness of the tube is reduced from 2.5 mm to 1.5 mm. The forming zone is gradually reduced. As shown in Figure 13, when the thickness-diameter ratio = 1.67%, the thickness is 2.5 mm. The critical wrinkling pressure is 7.8 MPa, and the critical rupture pressure is 15 MPa.

As shown in Figure 14, when the thickness-diameter ratio = 1.33%, the thickness is 2.0 mm. The critical wrinkling pressure is 10.3 MPa, and the critical rupture pressure is 13.6 MPa. Compared to Figure 13, the critical wrinkling pressure is increased by 32.1%, and the critical rupture pressure is reduced by 9.3%.

As shown in Figure 15, when the thickness-diameter ratio = 1.33%, the thickness is 2.0 mm. The critical wrinkling pressure is 11.0 MPa, and the critical rupture pressure is 11.9 MPa. Compared to Figure 13, the critical wrinkling pressure is increased by 41.1%, and the critical rupture pressure is reduced by 20.6%.

Therefore, for the component with a small thickness-diameter ratio, the process should be strictly controlled as a result of a narrower forming zone.

## 4. Experiments and Simulation Analysis

### 4.1. Experimental

The simulation and experimental equipment for TUBG are shown in Figure 16. The speed of the punch stroke is set to 2 mm/s with friction coefficient *μ* = 0.05 between tube and mode. When simulating, assume the extrusion model is isothermal. Both the mold and punch are set to a rigid body, and the tube is set to a deformed body. In this paper, the global seed method is selected to strictly control the grid density and determine the size of the grid to be 1 mm^2^.

Before the experiment, the shortened length in the axial direction was calculated. The calculation results showed that for tubes with a length of 200 mm, the shortened length in the axial direction is 8 mm. It means only 4 percent of the length is required in the axial direction during TUBG. Due to the small proportion, the length of this part is negligible in experiments.

Combined with the results of theoretical analysis, using the liquid pressure pump to supply internal pressure. The pump is connected to the tube. The tube is placed inside the mold. The geometry of the mold limits the minimum and maximum diameter of the nozzle. Through a catheter, the liquid in the pump can be injected into a rubber container, which in turn provides pressure. When TUBG begins, by controlling the movement of the punch, the tube is deformed under the force of punch and mold. By operating the pump, the pressure provided by the liquid drives the tube to have plastic deformation.

### 4.2. Plastic Deformation Process

#### 4.2.1. Simulation Accuracy

During the upsetting process of hollow tubes, it is very easy to shrink and wrinkle. As shown in Figure 17, four wrinkles appear around the edge when the upsetting is finished, with a low pressure of 9 MPa. Moreover, as the stamping process progresses, the wrinkle gradually evolves in size and becomes a dead wrinkle, which directly leads to the failure of the forming process.

At the same time, it is worth noting that, during the simulation and experimental forming process, the axial direction of the tube gradually evolves into annular wrinkles. As shown in Figure 18a,b, the emergence and evolution of this type of wrinkle. Figure 18c shows the wrinkle morphology after the experiment. It is considered to be the result of increased friction between the tube and mold. At the beginning of the forming process, the contact area between the tube and mold is small. The value of friction has not yet reached the critical value of wrinkling. When the contact area between the tube surface and the mold increases, the value of friction increases and leads to wrinkles eventually. Whether it is a beneficial wrinkle and whether it can be flattened in bugling remains to be discussed.

In this process, the extrusion force required per unit area can be obtained according to the empirical formula [33]:(34)$$P_{\alpha }=\frac{2}{\sqrt{3}}\bar{\sigma }\mu (1+C)\ln(\frac{R_{1}-R_{2}}{r_{1}-r_{2}})$$

*σ* is the average equivalent force value; *R*_1_, *R*_2_ are the outer diameter and inner diameter of the experimental tube; *r*_1_, *r*_2_ are the outer diameter and inner diameter of the thin wall of the hollow cone; μ is the coefficient of friction. When the friction increases, it is extremely difficult for the tube to complete a stamping process due to the press of internal pressure. The tube is more prone to wrinkling. It is extremely important to reduce friction between the tube and mold for the TUBG process.

#### 4.2.2. Forming Zone

When bulging, if the internal pressure is too low, it is difficult to make the tube expand to the predetermined shape, which cannot fit with the mold and meet the shape requirements. If the internal pressure is too high, not only better performance equipment is required, but also the tube may be ruptured, which results in the failure of nozzle manufacturing.

The materials selected for this experiment are 20 steel. The strength coefficient (*K*) is 800 MPa, hardening index (*n*) is 0.2, anisotropy (*R*) is 0.88, wall thickness is 2 mm, diameter is 75 mm, and length is 200 mm. Combined with the formula in this article, the forming zone is given, as shown in Figure 19. The initial well-formed upsetting process does not require internal pressure support. As the punch stroke increases, the required support internal pressure gradually appears and increases. Combine the support pressure with the rupture pressure, forming zone of TUBG in this experiment is given.

#### 4.2.3. Deformation Analysis

As shown in Figure 20, the process of TUBG is shown. The first stage of TUBG is shown in Figure 20a. The tube completes the well-formed upsetting process with the support of internal pressure. However, there are three wrinkles in the force transmission area. As the experiment progresses and pressure increases, as shown in Figure 20b, two wrinkles are flattened. Most parts of the tube have been fitted to the mold. Finally, as shown in Figure 20c, after the internal pressure increases to the set value, all wrinkles are flattened, and no rupture occurs. A well-formed rocket nozzle is obtained.

In order to explore the accuracy of the forming zone, the upper and lower limits of the forming zone are taken respectively. As shown in Figure 21a–c, when P = 13.9 MPa, the evolution process of tube bulging is discussed. With the increase of internal pressure, the wrinkles are flattened to a certain extent. Finally, after forming, the tube was perfectly molded.

In order to explore the forming results under the different pressure in the forming zone, this paper discusses the forming conditions under P = 11.9 MPa. As shown in Figure 21d–f, with the increase of internal pressure, the wrinkles are flattened to a certain extent. Finally, the tube fits into the mold and forms a well-formed part. It also indicates that the wrinkles formed under this coefficient of friction are beneficial.

For the rocket nozzle to be formed, its wall thickness shows a gradual thinning phenomenon from the small end to the large end. With the improvement of the uniformity of the wall thickness change, its ability to resist ablation during the working process is enhanced.

In this process, the theoretical calculation of Section 3 shows that the wall thickness is thickened after upsetting, and the wall thickness is thinned after bulging. However, in order to explore the wall thickness distribution under different pressure, the corresponding strain is extracted for analysis. As shown in Figure 22, compared to P = 11.9 MPa, the strain evolution of the bulging segment under 13.9 MPa is more uniform. For P = 13.9 MPa, considering that due to the greater pressure, the greater wrinkling behavior in the force transmission zone of the material occurs. After repeated action of the internal pressure bulging shape, an uneven strain distribution phenomenon occurs.

Therefore, when manufacturing nozzle parts, a process scheme with a larger value of P should be selected in the forming zone. Especially compared to spinning, which uses a steel plate to make the rocket nozzle, TUBG has a smaller degree of deformation. Based on this, many defects are avoided.

Furthermore, the strain distribution along the axial direction is extracted, as shown in Figure 23. In contrast, when P = 11.9 MPa, the real value is more fitted to the theoretical value. In the real strain distribution, the location of the mutation is the same as the wrinkle position during the simulation. It is considered that the appearance of beneficial wrinkles has caused the sudden change in the strain value there.

#### 4.2.4. Experimental Results

Figure 24 shows the outline of a rocket nozzle. For this component, the diameter ratio of small end to large end is 31.67%. When it is formed, the difficulty is that the small end diameter is 75 mm, and the large end diameter is 200 mm.

In the process of TUBG, under the pressure of 13.2 MPa, the rocket nozzle of the above size can be manufactured without any wrinkling and rupturing defects. As shown in Figure 25a-(1), with an internal pressure lower than the critical value, an imperfect upsetting process occurs. Wrinkles appear on the tube. As shown in Figure 25a-(2), within the given forming zone, the tube undergoes a well-formed TUBG process. A well-formed rocket nozzle is obtained. As shown in Figure 25a-(3), under the pressure above the critical value, annular wrinkles appear in the force transmission zone of the tube. It is considered that excessive internal pressure increases friction, which leads to the appearance of wrinkles. As shown in Figure 25b, the wall thickness of the perfect nozzle obtained in Path 2 is measured. The thickness values from the finite element analysis results are also extracted. From the small end to the big end of the rocket nozzle, the thickness shows a trend from thicker to thinner, which meets the requirements for the use of the rocket nozzle.

## 5. Conclusions

The traditional forming process of the rocket nozzle has the problems of multi-step forming, the occurrence of defects, and severe plastic deformation. This study proposed a novel tube upsetting-bulging (TUBG) method. Both theoretical and experimental works are conducted to analyze plastic deformation behavior. Detailed conclusions can be drawn as below:

1. A theoretical model for wrinkling and rupturing is built. Based on the energy method, a theoretical critical internal pressure is established, which can avoid wrinkling and rupturing. The model considered the factors such as materials and geometric parameters. For a certain degree of deformation, the critical pressure is improved with the increase of the strengthen coefficient and anisotropies but decreases as the hardening exponent increases. A theoretical forming zone for TUBE is established, which decreases with the decrease of the thickness-diameter ratio.

2. A finite element analysis model is set up. The accuracy of this model is verified by the wrinkling behaviors of different cross-sections. The plastic deformation behavior of TUBG is analyzed. During deformation, the small end is compressed and thickened. The large end is stretched and thinned. Wrinkles appear at the beginning of the deformation, which can unfold in the later stages. The finite element analysis results show that the strain distribution is consistent with the theoretical predictions.

3. An experimental platform is established, and verification is carried out. Three loading paths are selected near the forming zone for testing. When the pressure is less than the critical value, an imperfect upsetting process occurs with wrinkles appearing. When the loading path is inside the forming zone, a well-formed rocket nozzle can be obtained. When the pressure is higher than the critical value, axial wrinkles appear due to the increased friction force.

4. The study shows that TUBG combines the advantages of traditional schemes. Using the predicted forming zone, a well-formed nozzle can be obtained with high efficiency. The strain distribution of the specimens is relatively uniform.

## Figures and Tables

**Figure 1 materials-16-01680-f001:**
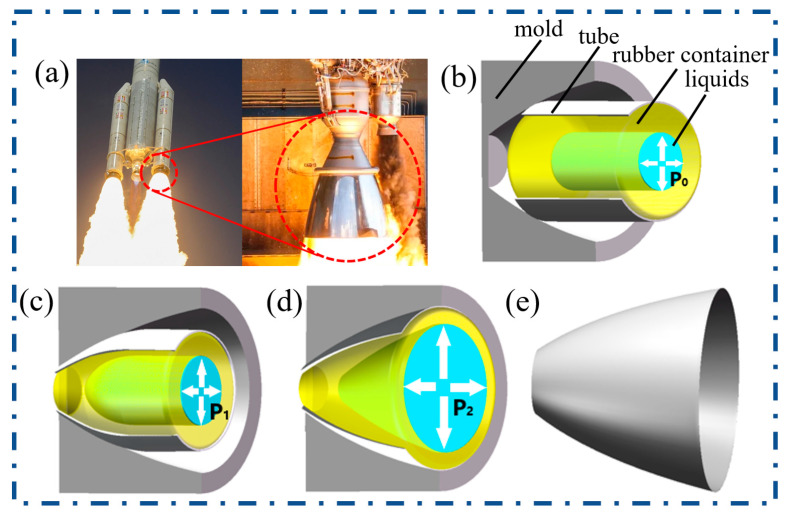
Nozzle of rocket and schematic of TUBG. (**a**) The rocket nozzle, (**b**) The schematic of TUBG with initial internal pressure P_0_, (**c**)The tube after upsetting with higher internal pressure P_1_, (**d**) The tube after bulging with much higher internal pressure P_2_, (**e**) The perfectly formed rocket nozzle.

**Figure 2 materials-16-01680-f002:**
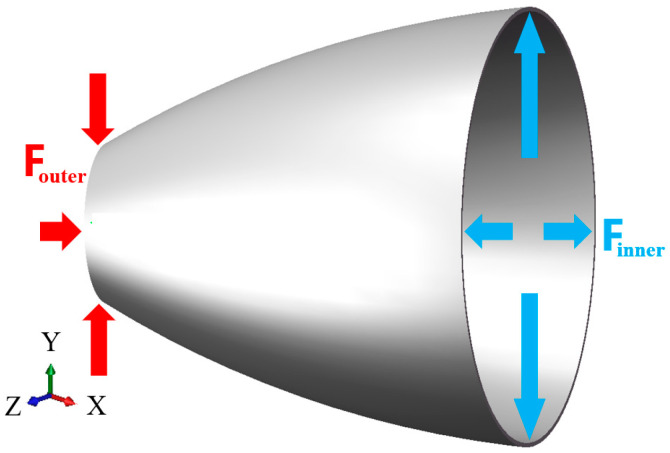
Outer and inner forces where lead wrinkles and ruptures.

**Figure 3 materials-16-01680-f003:**
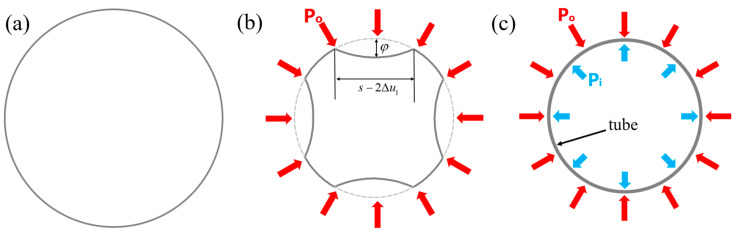
Cross-sectional topography of tube, (**a**) Initial tube, Tube after upsetting (**b**) with wrinkles and (**c**) without wrinkles (from the perspective of the XOZ plane).

**Figure 4 materials-16-01680-f004:**
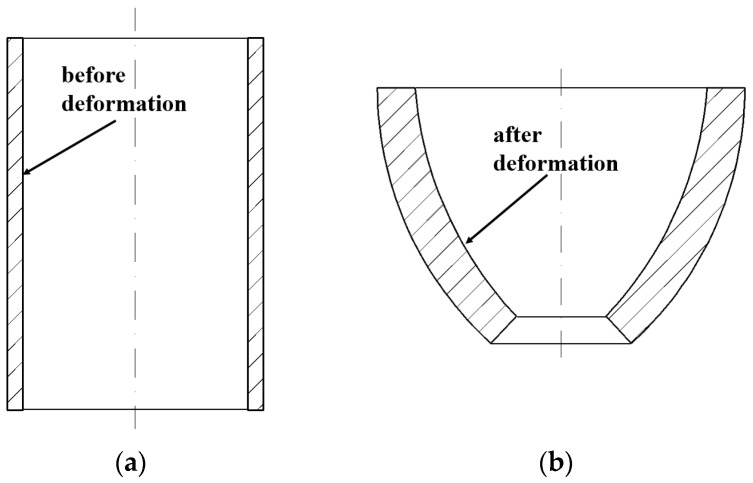
Schematic diagram of the tube (**a**) initial state and (**b**) after upsetting (from the perspective of the XOY plane).

**Figure 5 materials-16-01680-f005:**
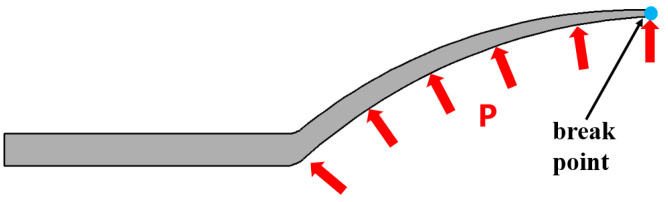
Schematic diagram of the bulging on the tube (from the perspective of the XOY plane).

**Figure 6 materials-16-01680-f006:**
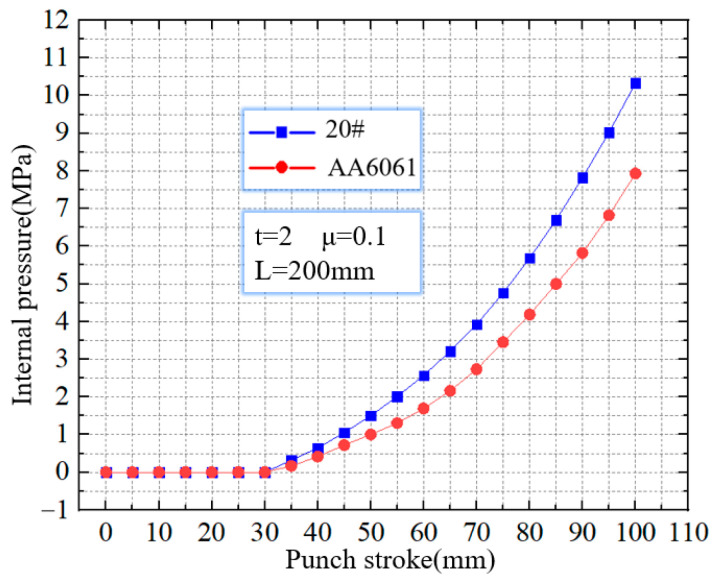
The critical wrinkling pressure of 20 steel and AA6061.

**Figure 7 materials-16-01680-f007:**
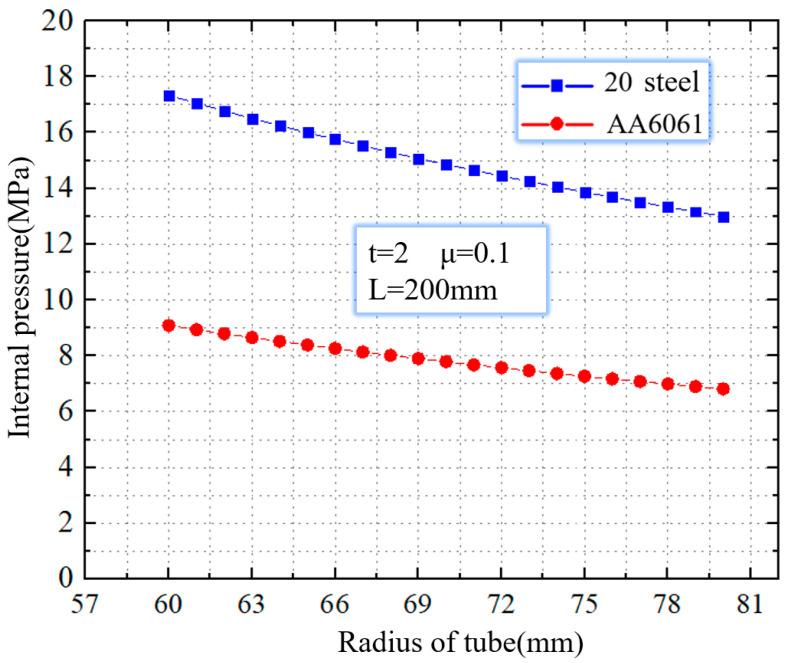
The critical rupture pressure of 20 steel and AA6061.

**Figure 8 materials-16-01680-f008:**
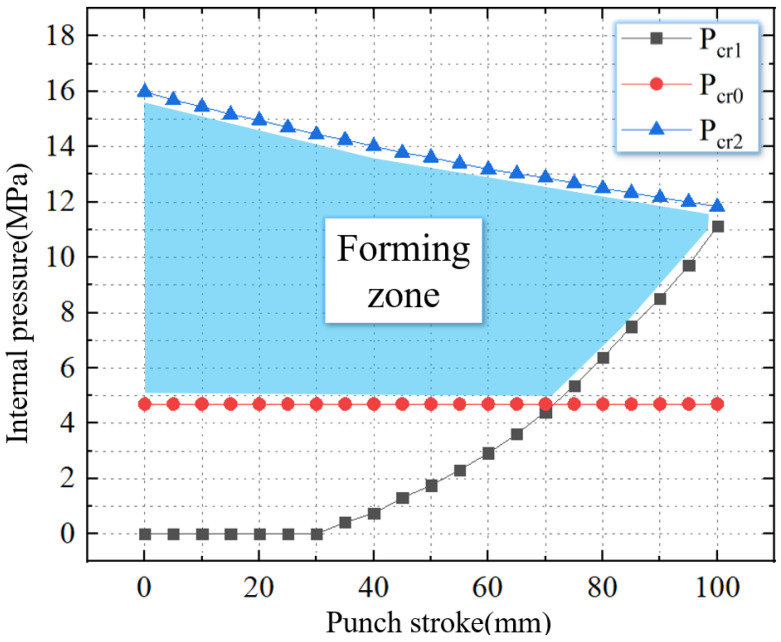
The forming zone of TUBG.

**Figure 9 materials-16-01680-f009:**
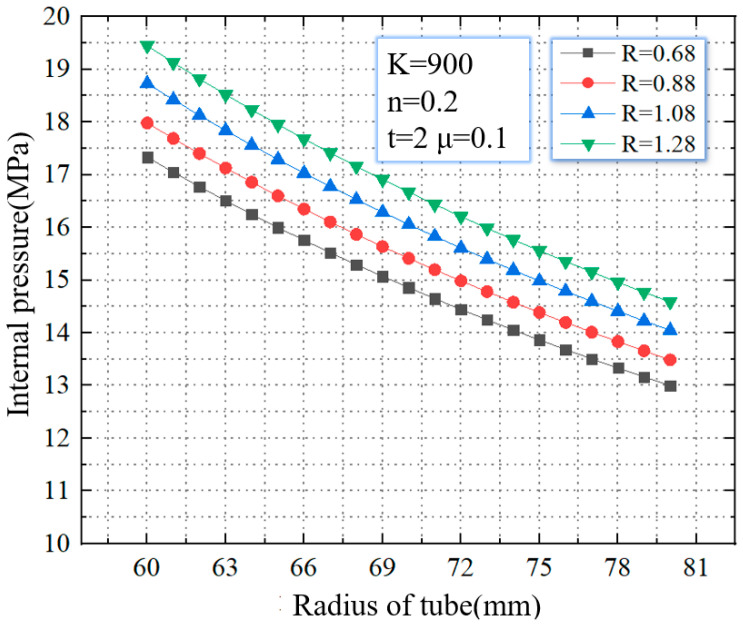
Critical internal pressure for rupture with different anisotropies.

**Figure 10 materials-16-01680-f010:**
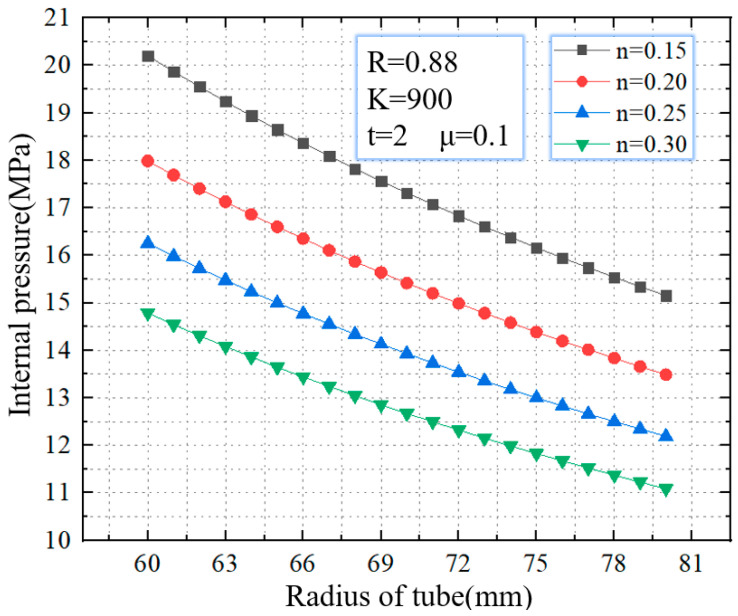
Critical internal pressure for rupture with different strain hardening exponents.

**Figure 11 materials-16-01680-f011:**
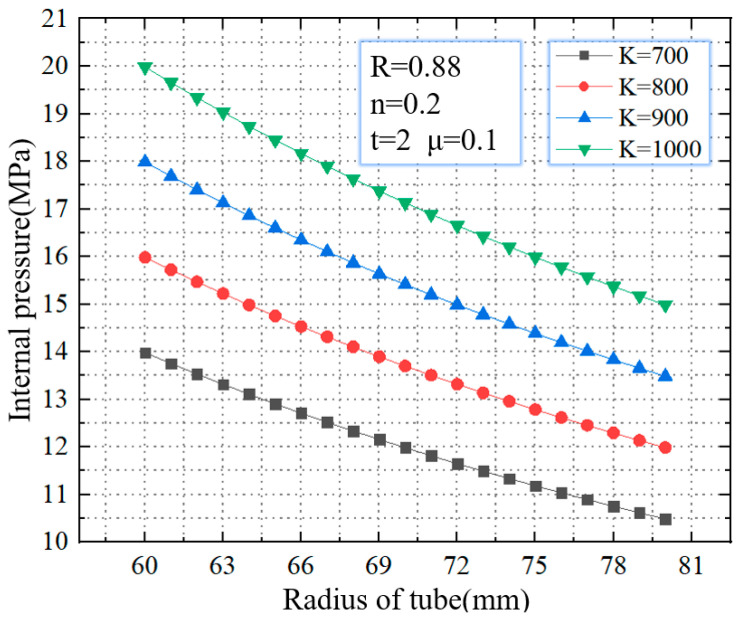
Critical internal pressure for rupture with different strength coefficients.

**Figure 12 materials-16-01680-f012:**
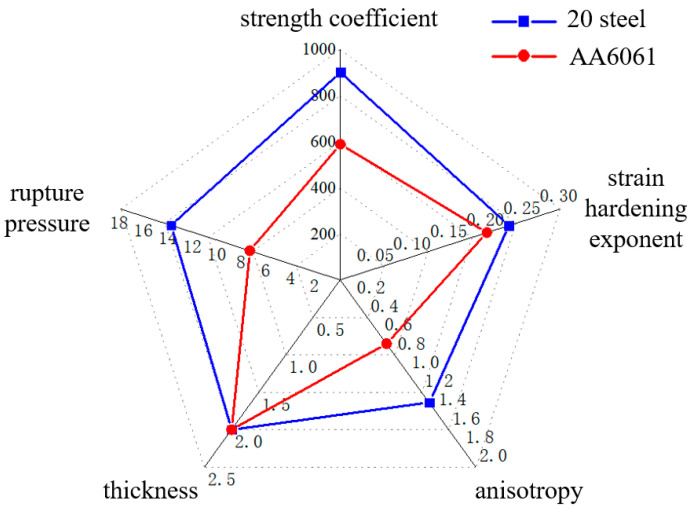
Comparison of critical rupture pressure between 20 steel and AA6061 using TUBG.

**Figure 13 materials-16-01680-f013:**
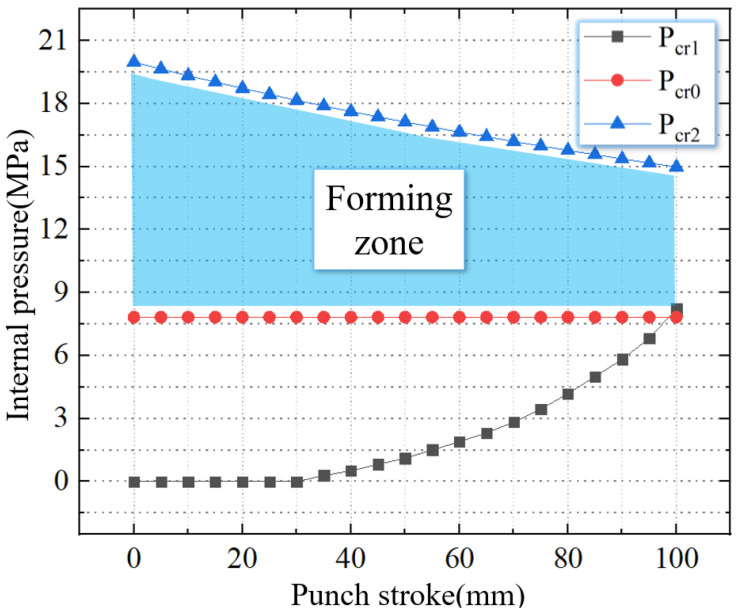
The forming zone of TUBG (thickness-diameter ratio = 1.67%).

**Figure 14 materials-16-01680-f014:**
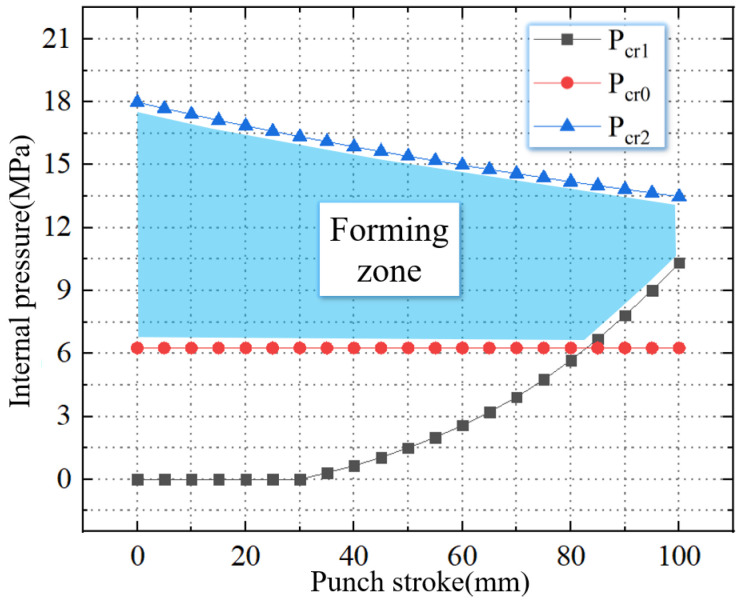
The forming zone of TUBG (thickness-diameter ratio = 1.33%).

**Figure 15 materials-16-01680-f015:**
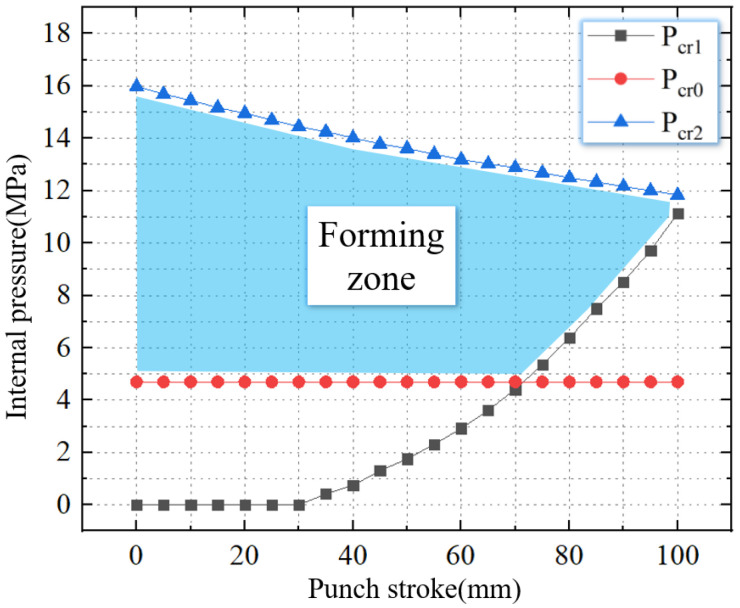
The forming zone of TUBG (thickness-diameter ratio = 1.00%).

**Figure 16 materials-16-01680-f016:**
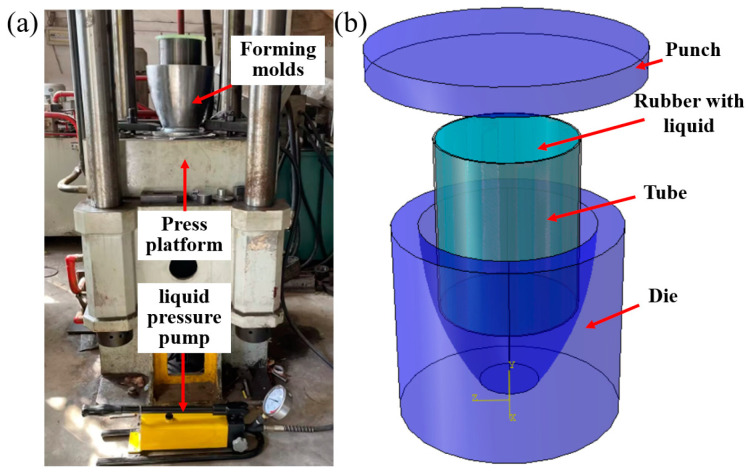
The equipment and simulation model for TUBG. (**a**) The equipment for TUBG, (**b**) The simulation model for TUBG.

**Figure 17 materials-16-01680-f017:**
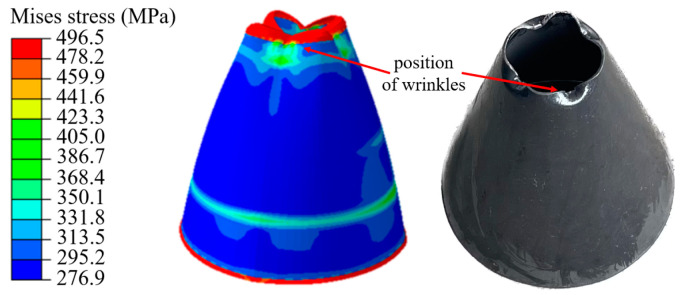
Wrinkle in XOZ plane during TUBG process with pressure of 9 MPa.

**Figure 18 materials-16-01680-f018:**
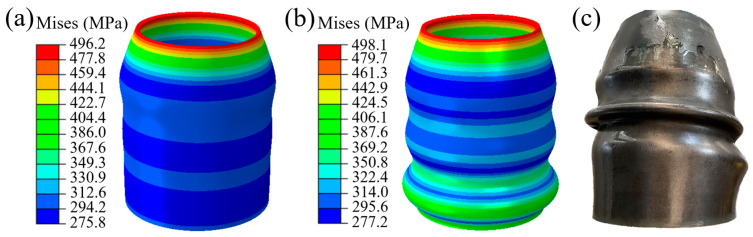
Wrinkles in XOY plane during TUBG process with a pressure of 20 MPa. (**a**) The emergence of wrinkle, (**b**) The evolution of the wrinkles, (**c**) The wrinkle morphology after the experiment.

**Figure 19 materials-16-01680-f019:**
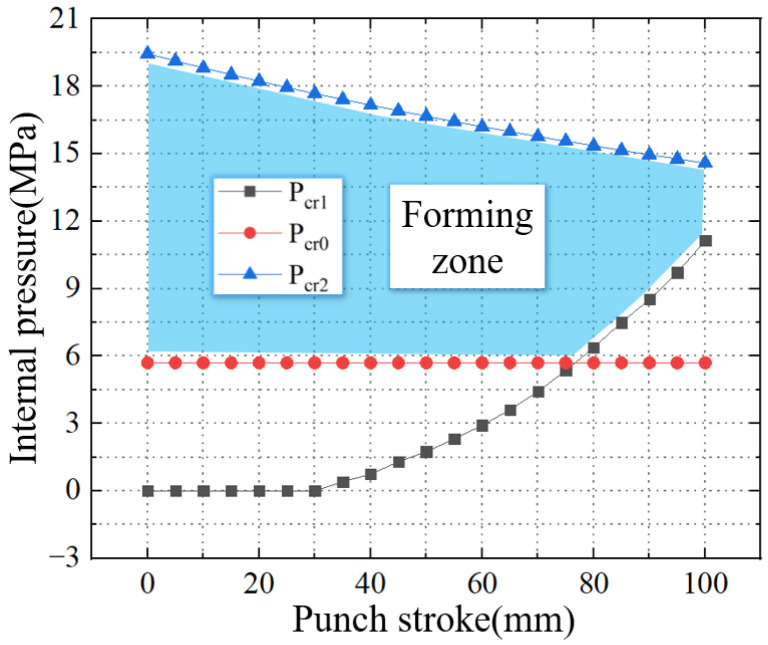
Forming zone of TUBG in this experiment.

**Figure 20 materials-16-01680-f020:**
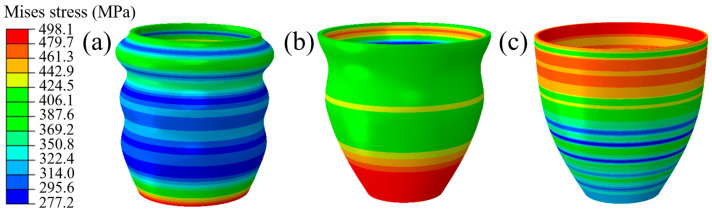
Mises stress distribution of formed parts using TUBG. (**a**) The first stage of TUBG, (**b**) The shape of the nozzle that two wrinkles are flattened, (**c**) A well-formed rocket nozzle.

**Figure 21 materials-16-01680-f021:**
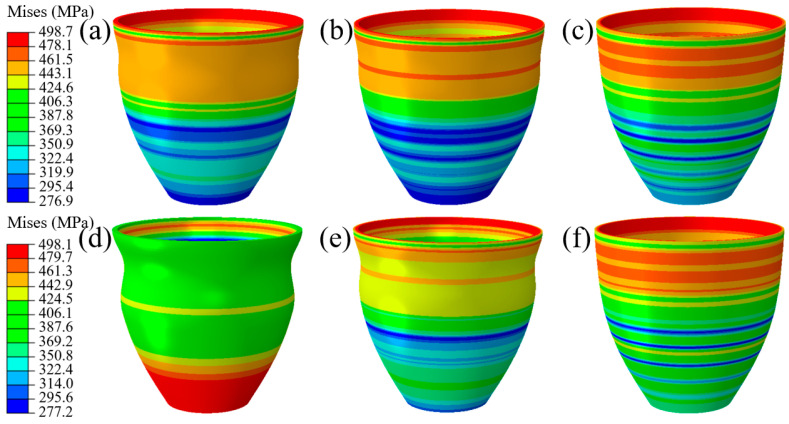
Mises stress distribution of formed parts in cases of 11.9 MPa and 13.9 MPa. (**a**–**c**) The evolution process of wrinkle morphology when P = 13.9 MPa (**d**–**f**) The evolution process of wrinkle morphology when P = 11.9 MPa.

**Figure 22 materials-16-01680-f022:**
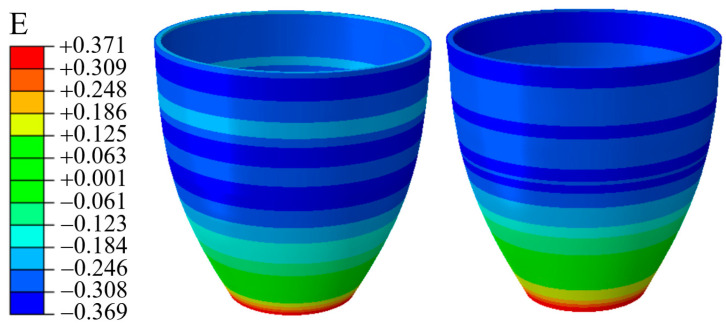
Strain distribution in cases of 11.9 MPa and 13.9 MPa.

**Figure 23 materials-16-01680-f023:**
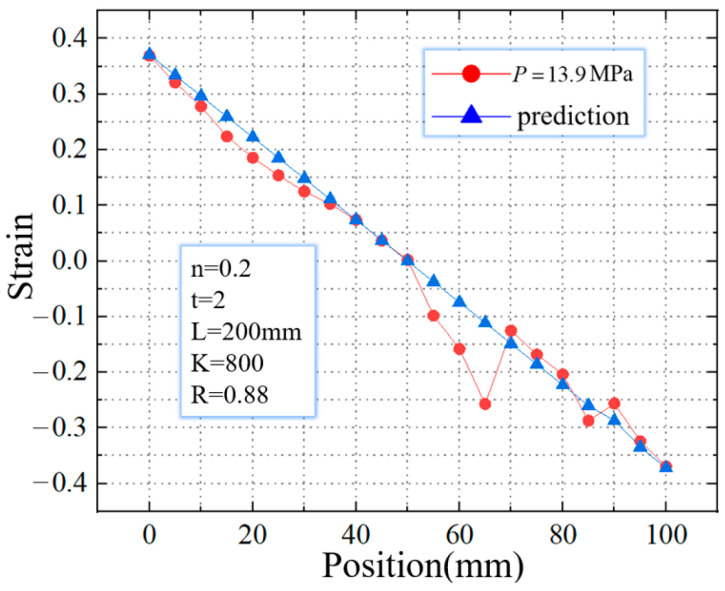
Comparison of strains between calculated prediction and simulation result.

**Figure 24 materials-16-01680-f024:**
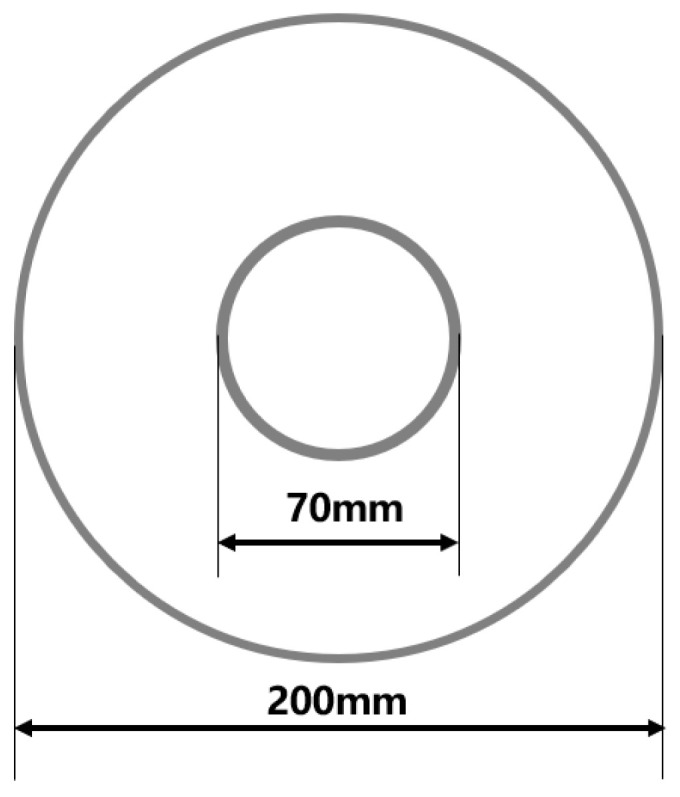
The diameter changes of the target part in this process scheme.

**Figure 25 materials-16-01680-f025:**
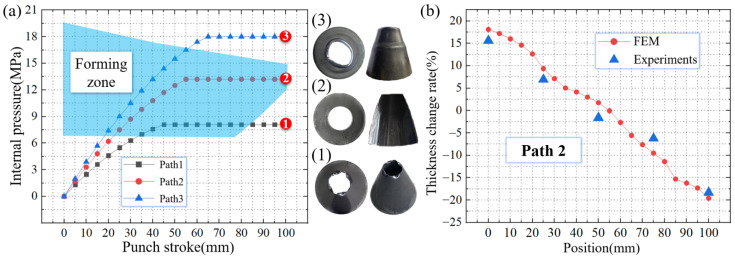
Rocket nozzle making by TUBG and its thickness value. (**a**)The forming zone and rocket nozzle making by TUBG, (**b**) The wall thickness of the perfect nozzle.

## Data Availability

The data used to support the findings of this study are available from the corresponding authors upon request.

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
