# Peer review of "Theoretical Analysis and Verification on Plastic Deformation Behavior of Rocket Nozzle Using a Novel Tube Upsetting-Bulging Method"

_materials, 2023, doi:10.3390/ma16041680_

Round 1
Reviewer 1 Report
The article is interesting. The authors need to address the attached comments before proceeding further.

Author Response
Many thanks for your comments on our manuscript entitled "Theoretical analysis and verification on plastic deformation behavior of rocket nozzle using a novel tube upsetting-bulging method". Many constructive suggestions which would help us in depth to improve the quality of the paper are given. All the changes are highlighted in yellow in the revised manuscript and the point-by-point responses to the comments of you are as follows. We hope the revised manuscript and the response have addressed your comments and suggestions well.
Please see the attachment.

Reviewer 2 Report
The authors discuss an improved method of manufacturing rocket nuzzles, using both theory and experimental methods. The research seems interesting enough to merit publication, but the overall presentation must be greatly improved:
On the one hand, many English grammar mistakes make it very difficult to follow, on the other hand numerous steps and underlying assumptions are left out making it near impossible to reproduce the theory derivation.
For example, half of the variables appearing on page 5 are not defined, in eq. (10) the authors do not mention that sigma and epsilon are effective values (in general stress and strain are tensors) for the special deformation under consideration (ref. [26] defines these quantities for the bulging plate), the assumption (small x?) leading to eq (26) is not mentioned, and several other steps/assumptions in the derivations of this section are missing.
Author Response

(The authors gave the same response as above.)

Reviewer 3 Report
I considered the manuscript entitled; “Theoretical analysis and verification on plastic deformation be-2 havior of rocket nozzle using a novel tube upsetting-bulging 3 method”. The manuscript is well organized and written, however, there are some ambiguities such as:
1- After equation (4) it is written “In which m is the frequency” but m is not seen in the equation
2- How are equations 4 to 6 derived?
3- The brackets and dimension of equation (21) should be checked.
4- The dimension of equation (22) and (24) should be checked.
5- How is pressure applied? It is suggested that schematic diagram for sealing the process is shown.
Due to the large number of notations in the equations, it is suggested to prepare a nomenclature.
Author Response

(The authors gave the same response as above.)

Round 2
Reviewer 1 Report
The authors have incorporated the comments and suggestions raised by the reviewer and the manuscript is now acceptable for publication
Author Response
Thanks for your reply. We would like to appreciate you for your work on our manuscript.
Reviewer 2 Report
The authors have added many details which certainly improve the paper somewhat, but the English is still very bad, making it hard to follow at times. I strongly recommend using a professional editing service.
Author Response
Thank you again for your suggestions for our article. We would like to appreciate you for your work on our manuscript. We have double-checked the manuscript carefully and corrected mistakes timely. With your valuable suggestions, we improved the quality of the language. The detailed variation can be found in this file. All the changes in the second round are highlighted in yellow in the revised manuscript. We hope the revised manuscript and the response have addressed your comments and suggestions well.
Please see the attachment.

Reviewer 3 Report
The dimensions of the mathematical expressions in equation 21 are problematic. The first mathematical expression in the bracket has the dimension MPa, while the second expression has the dimension MPa.m. Please check and correct the equation. The Rr/R0 expression caused the problem. Because it should be dimensionless, which it is not.
Author Response
Thank you again for your suggestions for our article. We would like to appreciate you for your work on our manuscript. We have double-checked the manuscript carefully and corrected mistakes timely. Also, we improved the quality of the language. The detailed variation can be found in this file. All the changes in the second round are highlighted in yellow in the revised manuscript. We hope the revised manuscript and the response have addressed your comments and suggestions well.
Please see the attachment.
